

# A comprehensive data quality evaluation method for the current of marine controlled-source electromagnetic transmitter based on Analytic Hierarchy Process

Rui Yang[1], Meng Wang[1], Gongxiang Wang[2], Ming Deng[1], Jianen Jing[1], Xiancheng Li[1]

[1]China University of Geosciences (Beijing), Beijing, 100083, China
[2]Guangzhou Marine Geological Survey, Guangzhou, 510075, China

*Correspondence to*: Meng Wang (wangmengcugb@qq.com)

**Abstract.** Marine controlled-source electromagnetic method has more and more applications in ocean resources exploration. Electromagnetic transmitter sends electromagnetic wave to the underground, the receiver located on the seafloor receives the

10 electromagnetic wave which carries the information of the geosphere. And the underground structure is obtained by inversion calculation. Data quality of electromagnetic transmitter and seafloor receivers is the most important part of this method. The quality level of transmitting current directly affects the signal-to-noise ratio (SNR) of the electromagnetic field data, as received by a multi-component electromagnetic receiver from the seabed. Although the transmitting current stability is sufficient under normal circumstances, the SNR of the received signal can change owing to factors such as outside noise.

In some emergency cases such as instrument failure or a sudden increase in electromagnetic interference that we are not aware of, the frequency and properties of the transmitting current may change, such as its size and waveform. The traditional current monitoring and data playback tools fail to detect and evaluate the anomalies well and in a timely manner, which introduces considerable errors in the later data processing procedure. Pertaining to these issues, this paper proposes a comprehensive quality evaluation method for the transmitting current. The proposed algorithm, based on the analytic

hierarchy process, is first used to analyse five current stability parameters: current frequency, positive amplitudes, negative amplitudes, discrepancy of ideal waveform, and waveform repetition and then to define the harmonic energy and calculate the quality of transmitting current (QTC) index of the final data to assess the quality of the transmitting current comprehensively. The results of a marine experiment performed in 2016 show that the algorithm can identify abnormal current data and quantitatively evaluate the current conditions. Under normal circumstances, the QTC index is within 2%.

However, after the simulation of anomalous mutations of the various attributes, the QTC index synchronized mutations to more than 4% and some curvilinear features were observed. These results will provide a positive, significant guide for the evaluation and monitoring of transmitting current data in marine experiments.

**Keywords:** Marine controlled-source electromagnetic, Quality of transmitting current, Analytic hierarchy process, QTC

index, Signal-to-noise ratio



## 1 Introduction

The marine controlled-source electromagnetic (MCSEM) technique is an effective method of exploring natural gas hydrate reservoirs and petroleum reservoirs (Constable and Srnka, 2007, Wang et al., 2013). With the development of the MCSEM method worldwide, it not only has been used to develop a series of algorithms for forward and inverse calculations (Gribenko et al., 2007; Jing et al., 2016), but also has shown great potential for use in practical applications (Cox et al., 1986, Constable, 2010). In realistic marine prospecting work, an MCSEM exploration system is usually composed of a high-power, controlled-source electromagnetic transmitter and a submarine mixed field source electromagnetic receiver (Chen et al., 2017a and 2017b, Di et al., 2018). Each component remarkably influences the quality, precision, and interference of the electromagnetic field signals. Ensuring the high quality and high signal-to-noise ratio (SNR) of the transmitting current are the most important tasks of the transmitting system. In 2007, EMGS (Electromagnetic Geoservices) company studied the waveform of a transmitting current from the perspective of harmonic energy ratio (HER) and proposed a transmitting current waveform that improves the signal quality of the electromagnetic field (Mittet et al., 2008). Currently, there is no quantitative method available for the reliable evaluation of the transmitting current data in actual MCSEM work (Edwards, 2005, He et al., 2009, Luan et al., 2018). In this paper, we propose a comprehensive quality evaluation algorithm based on the analytic hierarchy process (Saaty, 1987) for the transmitting current. Considering the impacts of factors such as the frequency stability, positive and negative amplitude stability, discrepancy from the ideal waveform, waveform repetition, and HER, as well as the distribution weight value, a quality of transmitting current (QTC) index can be calculated. This research also provides a reliable path for transmitting current improvement and realization of the real-time monitoring systems.

## 2 Transmitting current analysis

A personal computer (PC) in a deck unit, used for the real-time monitoring of the transmitting current, monitored the changes in the current amplitude based on the simple qualitative observations. Figure 1 shows basic parameters such as the current waveform and frequency components of channel 1 (we used multiple channels to record the data from multiple current sensors). The quality and stability of the transmitting current depend on various parameters such as the transmitting frequency, amplitude, difference from the ideal amplitude, degree of waveform repetition, and harmonic energy distribution. The MCSEM operation data pre-processing and inversion considerably influence the transmitting current quality and reliability. Therefore, the comprehensive evaluation and feedback of the transmitting current data quality are particularly important.


## 3 Evaluation algorithm

Earlier, PCs on decks performed real-time monitoring of transmitting current parameters that was only limited to observing changes in amplitude, a quality that can only be described qualitatively. To evaluate the properties of current quantitatively, more parameters must be calculated, predominantly the actual work factors such as the frequency, amplitude, ideal value difference, waveform repetition, and harmonic energy multiple aspects that influence the quality of the received data. In order to facilitate the analysis of fast Fourier transform (FFT) operation, the data is divided into blocks by fixed-period number(N) and the corresponding length of time(T) can be obtained by dividing N by sampling rate(fs) as shown in equation 1:

$$N = 2^n$$
$$T = N/fs \,, \tag{1}$$

where n is a positive integer, its value depends on the length of the original data (ensure that there are enough data blocks for analysis) and the transmission frequency($T \gg 1/f$) to be able to analyse the signals of more cycles. And after that, each data block is calculated by the algorithm. Then, one by one block analysis of the transmitting data is performed to calculate the QTC index, which reflects the current quality, to evaluate the transmitting current quantitatively.

## 3.1 Frequency stability

The transmitting source frequency is one of the core parameters of MCSEM and considerably impacts the solution process in the frequency domain. Therefore, the evaluation of the transmitting current must introduce frequency-related parameters. Considering the actual work parameters affecting the stability of electromagnetic waves, a frequency stability parameter ($a_i$) was defined and calculated as shown in equation 2:

$$a_i = n \left( x_i - \frac{1}{n} \sum_{j=1}^{n} x_j \right) \Big/ \sum_{j=1}^{n} x_j \,, \tag{2}$$

where $x_i$ is the frequency of present data block, $x_j$ is the frequency of each block used to calculate the frequency base value (average value), and n is the number of data blocks. The frequency axis is discretized (with step size is fs/N) when the fast Fourier transform is performed: hence, when an $x_i$ is input, the actual frequency can be determined by searching the local maximum amplitude. The stability is measured as the ratio of the actual frequency of each data block to the average frequency of all data blocks.

## 3.2 Positive amplitude stability

The MCSEM method is required to output an ac signal of a certain size while maintaining the stability of the actual transmitting current. Considering an actual transmitting current that deviates slightly numerically, reflecting the positive and negative power supplies of different circuits, the average sizes of the forward and reverse currents, defined as positive and negative amplitudes, respectively, and two parameters, positive and negative amplitude stability, were defined. The formulae for the positive amplitude stability are shown in equation 3:



$$b_1 = 0.001 \ (i = 0)$$

$$b_i = \frac{J_i^+ - J_{i-1}^+}{J_{i-1}^+} \ (i > 1) \ , \tag{3}$$

where $b_i$ is the positive amplitude stability of each data block and $J_i^+$ is the mean value of the positive sequence current data of data block i.

### 3.3 Negative amplitude stability

Similarly, the negative amplitude stability is given by equation 4:

$$c_1 = 0.001 \ (i = 0)$$

$$c_i = \frac{J_i^- - J_{i-1}^-}{J_{i-1}^-} \ (i > 1) \ , \tag{4}$$

where $c_i$ is the negative amplitude stability of each data block, $J_i^-$ is the average negative sequence current data of data block i. To evaluate a weighted combination later, the negative sequence is calculated as an absolute value.

### 3.4 Ideal waveform difference

In general, an MCSEM electric dipole transmitting source is a single frequency square wave signal or a mixed frequency signal. However, during an actual transmission, the electromagnetic wave is usually not a standard square wave, for various reasons. Based on this difference, the ideal waveform difference parameter can be defined as shown in equation 5:

$$d_i = \frac{1}{n}\sum_{k=1}^{n} \frac{J_k - J_{ki}}{J_{ki}} \ , \tag{5}$$

where $J_k$ is the kth value of current data block of the transmitting current, $J_{ki}$ is the ideal transmitting current corresponding to
$J_k$, and n is the quantity of data of this block.

### 3.5 Waveform repetition

The difference between the transmitting current waveforms in adjacent periods can be used to map the stability of the current data with time, and this degree of waveform variation is quantified as the waveform repetition degree parameter, which is given by equation 6:

$$e_i = \frac{1}{n-b}\sum_{k=b+1}^{n} \frac{J_k - J_{k-b}}{J_{k-b}} \ , \tag{6}$$

where $J_k$ is the kth value of the transmitting current data block, $J_{k-b}$ is the (k-b)th value, n is the number of data to be analysed, and b is the number of each cycle of the transmitting waveform.



### 3.6 HER stability

When frequency domain analysis is performed, the FFT tends to produce higher harmonics, which divide the energy to weaken the signal of fundamental frequency. To introduce a parameter that reflects harmonic energy, the stability of HER was defined, as shown in equation 7:

$$hr = \frac{1}{n}\sum_{i=1}^{n}(1 - hr_i),\qquad(7)$$

where $hr_i$ is the ratio of the actual assigned current to the theoretical current of the ith fundamental frequency point, which can be obtained by the conversion of frequency domain amplitude after the FFT transformation and n is the number of fundamental frequencies used for synthesis. In this sea trial, the two synthetic fundamental frequency points used were 0.5 Hz and 1.5 Hz, respectively, for n = 2.

### 3.7 Evaluation algorithm and comprehensive index

The above parameters are the factors that determine the quality of the transmitting current data. To evaluate the transmitting current quality comprehensively and quantitatively, five general data vectors, $a_i$, $b_i$, $c_i$, $d_i$, and $e_i$, are unified using the analytic hierarchy process (AHP) in the proposed method. Subsequently, the HER stability of all of the data (hr) is combined with w6 (an experience weight value) to obtain the composite QTC index. The flowchart of the algorithm is shown in Fig. 2(a). The algorithm can be applied if the QTC index 1) can show the difference between the stable and unstable transmitting currents and 2) has a certain ability to detect changes in various stability parameters.

The AHP is a structured technique for organizing and analyzing complex decisions, based on mathematics and psychology. It was developed by Saaty in the 1970s and has been extensively studied and refined since then. It is mainly used for subjective decision-making problems under the influence of multiple impact factors. It can be simply divided into the following steps: 1) building a structural model; 2) ranking the importance of the impact factors; 3) comparing and establishing a judgment matrix; 4) calculating the eigenvalues and eigenvectors of the judgment matrix; 5) calculating whether the consistency ratio CR is less than 0.10 with the maximum eigenvalue; if so, continue, otherwise return to step 2) and reorder the sorting; and 6) normalizing the eigenvectors of the largest eigenvalues to obtain the weight vector.

The hierarchical analysis model of the five aforementioned parameters is shown in Fig. 2(b). The measurement layer includes different channels, and each channel represents data recorded by different sensors. We only had data from two different sensors in this sea trial, but more channels including voltage and other parameters can also use this algorithm in the future. The rule layer contains five general parameters, and the index calculated by the AHP is in the target layer. The judgment matrix, presented in Table 1, is obtained by comparing the five parameter vectors using pairwise comparison of the degree of affecting current mass. For example, if b is the parameter we care the most about and d is the parameter we think has the least impact on current, then b/d = 9, d/b = 1/9; if e is more important than d, then b/e = 7, d/e = 1/2, and so on.

Consistency test: the consistency index CI is calculated and compared with the random consistency index RI to obtain CR. The formula for CI is



$$CI = \frac{\lambda_{max} - n}{n-1}, \tag{8}$$

where $\lambda_{max}$ is the largest eigenvalue of the judgment matrix and n is the number of factors.

RI is calculated by randomly constructing a sample matrix, presented in Table 2. The data in this table are the reference data that Saaty obtained through numerous random experiments and can be directly used to calculate CR.

Subsequently, CR can be calculated as shown in equation 9.

$$CR = \frac{CI}{RI}, \tag{9}$$

When CR < 0.10, the consistency of the judgment matrix is considered acceptable; otherwise, the judgment matrix should be modified appropriately.

When the CR condition is satisfied, the eigenvector corresponding to the maximum eigenvalue of the judgment matrix is

10 obtained. The vector obtained after normalization is the ranking weight ($\vec{w}$) of the relative importance of the corresponding factors at the same level as a factor in the previous level. The final QTC index of the ith data block is given by equation 10:

$$QTC_i = a_i w_1 + b_i w_2 + c_i w_3 + d_i w_4 + e_i w_5 + h_r w_6 , \tag{10}$$

where $\vec{w} = (w_1, w_2, w_3, w_4, w_5)$ is the corresponding weight vector of five indicators: frequency stability ($a_i$), positive amplitude stability ($b_i$), negative amplitude stability ($c_i$), ideal waveform difference ($d_i$), and waveform repetition ($e_i$): hr is

15 the HER stability; $w_6$ is an experience weight; and this formula is calculated in the time domain.

**4 Sea trial data evaluation**

Data examined in this study were obtained from the results of South China sea trials conducted in 2016. The analysis results are shown in Fig. 3. As can be seen from Fig. 3(a), the quality of the frequency stability data is high; the positive and negative amplitude changes are less volatile, mostly stable at 1%; and the ideal waveform difference is about 4% because the

20 actual transmitting current dose not reach the ideal value of 300 A (instead having a maximum of about 290 A). The main effect of it on QTC is to lower the average value. The waveform repetition represents the main reaction of the cycle stability of the output current waveform. And an anomaly occurs when the transmitting current is about to be turned off. It can be seen from Fig. 3(b) that the results are acceptable and the final QTC index is stable at 1%. The algorithm was developed for a single frequency, but the transmitting current in the CSEM method consists of multiple frequencies. To study whether the

25 QTC index was more stable at the component frequencies, the QTC index is calculated multiple times according to different frequencies and a spectrum was plotted. The spectrum diagram [Fig. 3(c)] mainly shows the changes in the QTC index with frequency and time. Blue indicates smaller stability values and more stable transmitting current, whereas red indicates the opposite. It can be seen that the main transmission frequencies of 0.5 Hz, 1.5 Hz, and higher-order harmonics are relatively stable. Hence, the overall results show that the higher frequencies are more stable than the lower frequencies.





## 5 Data mutation simulation and algorithm validation

To confirm the validity of the algorithm and the QTC index of each transmitting current stability parameter detection function, different attributes of square wave signals were considered as abnormal waiting to inspect the normal current data. By comparing the differences between the original data and the data after adding abnormal noise, the monitoring current was
simulated to analyze the data quality in actual operation.

### 5.1 Simulation of frequency mutation

To simulate the current data fluctuations caused by frequency mutations, a square wave signal with the same sampling rate and peak value as the original data was introduced with an analog signal of frequency 2 Hz. The signal was generated using computer code to simulate the frequency-varying signals and compared with the original signal to study the ability of the
QTC index to detect abnormal frequencies, as shown in Fig. 4. It can be seen from the spectrum analysis results that the amplitude of the data introduced in the signal segment increases significantly. Owing to the influence of odd harmonics, the current size of the frequency point of 6 Hz also increases. Figure 5 shows the changes in the original and analog signal QTC index curve (in blue and red, respectively). It can be seen that the QTC index with the mutation frequency band has noticeable changes of about 3%–4%. Therefore, the current QTC index can distinctly recognize frequency mutations.

### 5.2 Simulation of amplitude mutation

Current amplitude variation was simulated to test the ability of the algorithm to detect transmitting current mutations. The middle part of the data was selected for post-stack noise processing of the original data (Fig. 6). As the amplitude of only the noise segment is expanded, the overall Fourier transform results show only a slight increase in the amplitude of the base frequency point. In Fig. 7, the blue and red curves represent the QTC indices of the original signal and the data after adding
noise, respectively. It is apparent that there are considerable abnormal mutation points at the beginning and end of the noise section. Thus, the algorithm can well recognize current amplitude mutations.

### 5.3 Simulation of ideal waveform difference mutation

The difference between the actual and ideal current data waveforms is an important factor that affects the transmitting current quality. In the introduced noise section, a 0.5 Hz sine wave was generated to simulate the ability to detect differences
from the ideal square wave. It can be observed from the results in Fig. 8 and Fig. 9 that when the current waveform changes, the QTC index also changes, and the simulation data (red) decreases by 2%–3% compared to the original data (blue). Consequently, the waveform mutation certainly affects the actual QTC index.





### 5.4 Simulation of waveform repetition mutation

Square wave signals with various frequencies were used as noise sources to simulate waveform repetition degree mutation. The results are shown in Fig. 10 and Fig. 11. The data from three cycle transformations of square waves with frequencies of 0.5 Hz, 2 Hz, and 5 Hz, such that each waveform was continuously changing for the previous waveform, were used to 5 simulate the transverse mutations over time of actual transmitting waveforms. Since the QTC index of the simulated data is smaller than that of the original data, the algorithm also has a certain ability to recognize waveform repetition mutations.

### 5.5 Simulation of HER mutation

We defined the HER as the ratio of the energy of all the harmonics, except the fundamental wave, to the total energy. The smaller this value, the larger the expected fundamental wave energy and the base frequency SNR of the obtained data. A 10 high-frequency signal was simulated with an 8 Hz frequency square wave as noise, and when the fundamental frequency harmonic ratio (0.5 Hz, 1.5 Hz) energy declined, the harmonic energy increased. The original and simulated data are compared as shown in Fig. 12. It can be seen that the HER of simulated data decreases by 15%–20%. The QTC index of the simulated data (red curve in Fig. 13) increases by about 1% on average compared to the that of the original data (blue curve in Fig. 13). Hence, it can be concluded that the harmonic energy affects the QTC index.

### 15 5.6 Verification of measured data

The simulation results show that the QTC index has a good mutation identification effect on multiple parameters. In particular, it can identify abnormal data and reasons of the transmitting current for transmitting current quality evaluation and monitoring.

Figure 14 shows the transmitting current data from a test. In this experiment, a single-frequency (8 Hz) square wave signal 20 and mixed frequency (0.5 Hz, 1.5 Hz, 2.5 Hz) signals were sent. The current data in the first three sections of the current in this figure comprise the square wave signal, and the last part is the mixed frequency signal. The results obtained after calculation using our algorithm are shown in Fig. 15. In normal stable cases, the QTC index is around 1%. However, it mutates to 4%–5% when a large current signal is transmitted, and then returns to the normal value. Since a large current signal input is equivalent to a mutation signal and when the algorithm is used to identify the data changes, the QTC index 25 immediately reflects the corresponding mutations. Consequently, the algorithm can effectively identify transmitting current data anomalies.

### 6 Conclusion

In this paper, an algorithm was proposed to address the lack of a quantitative and comprehensive means of MCSEM transmitting current quality evaluation. After performing calculations using our algorithm, the QTC index can be obtained by



combining the frequency, amplitude, waveform difference degree, waveform repetition degree, and harmonic energy parameters of the transmitting current. The algorithm has the following characteristics:

1) The calculation process of the algorithm is stable and reliable. It can quantitatively calculate and evaluate the quality of the transmitting current data. Under normal circumstances, the QTC index stability is about 1%. However, when the current

frequency or other properties change, the QTC index increases to more than 4%. Hence, QTC indices less than 2% are normal and those more than 4% indicate current data exceptions that require correction in a timely manner.

2) The algorithm can detect the relationship between transmission current stability and current frequency. The QTC index of the fundamental frequency is smaller than those of the harmonic frequencies, and the frequency is negatively correlated with the QTC index. In other words, the fundamental frequency current and relatively high frequency current are more stable.

3) In addition to enabling quantitative transmission current quality evaluation, the algorithm can also detect abnormal situations in time and the reasons for an abnormal transmitting current. It can be seen from the results of the QTC index simulation of six parameters that an abnormal change in each parameter due to mutation certainly influences the composite index. The proposed algorithm can identify the reasons that these abnormal factors affect the transmission current quality. It can also provide a reference for real-time current monitoring of multiple attributes in future research.

**Data availability.**   There are no publicly available data for this study.

**Author contributions.**   Rui Yang wrote the code for the research and analyzed the results. Meng Wang provided the data required for the experiment and research ideas. Gongxiang Wang provided a lot of help in the experiment. All authors read

and provide written comments on this manuscript.

**Competing interests.**   The authors declare that they have no conflict of interest.

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

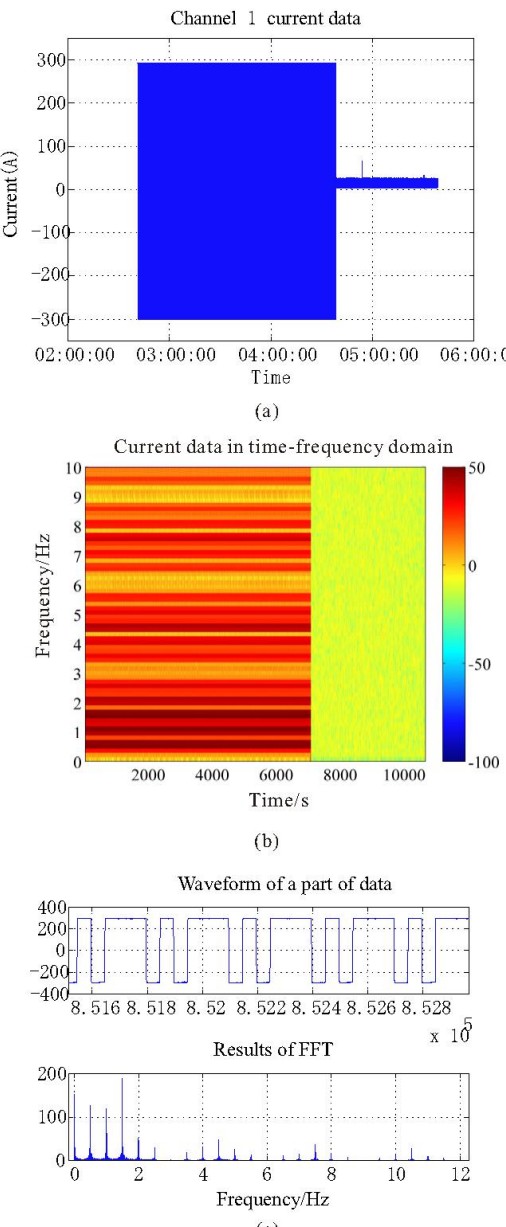

**Figure 1: (a) Transmitting current data, (b) time-frequency spectrum diagram, and (c) waveform diagram intercepted during working in the South China sea in 2016, with a sampling rate of 150 Hz.**



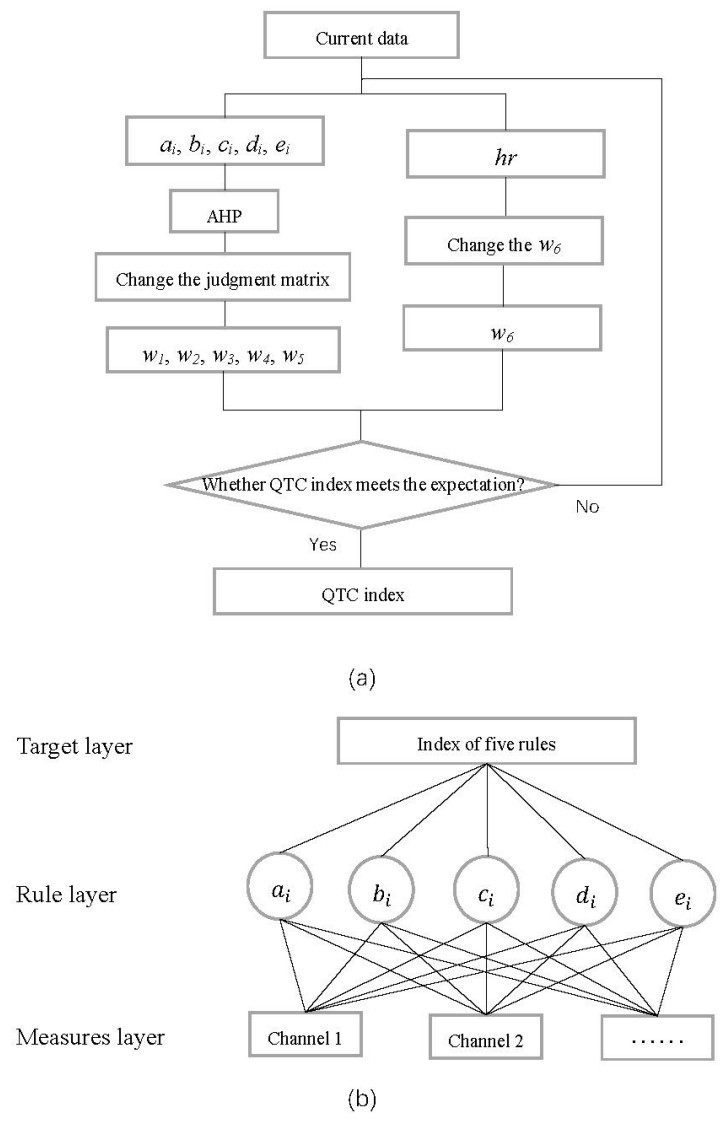

Figure 2: (a) Flow diagram of the algorithm and (b) AHP model of five parameters.





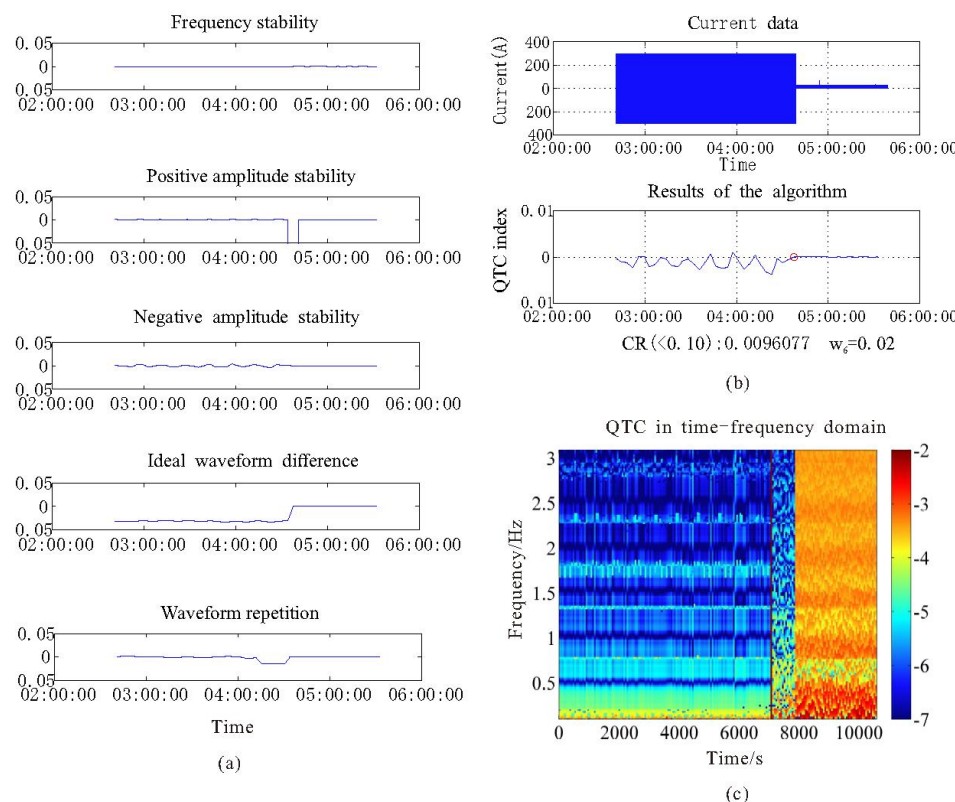

**Figure 3: Comprehensive analysis results of sea trial data in 2016.**

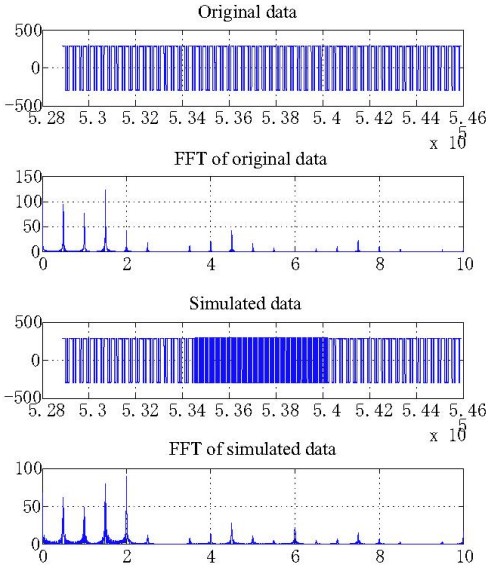





**Figure 4: Original data(above)and simulated data of frequency mutation(below).**

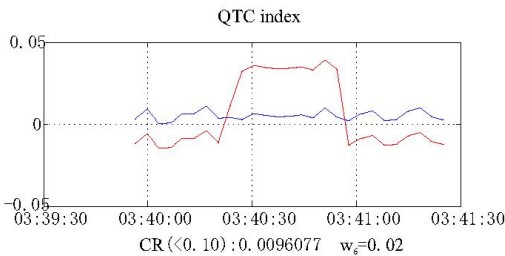

**Figure 5: QTC index of original data(blue)and simulated data of frequency mutation(red).**

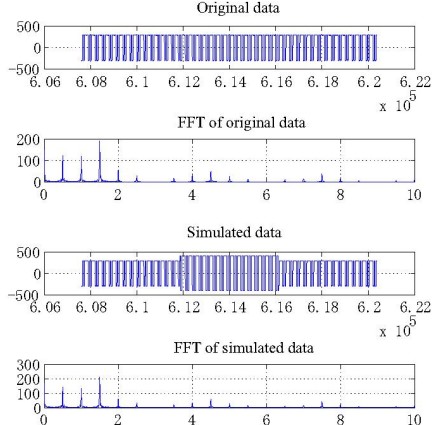

5 **Figure 6: Original data(above)and simulated data of amplitude mutation(below).**

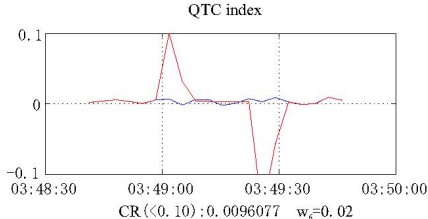

**Figure 7: QTC index of original data(blue)and simulated data of amplitude mutation(red).**





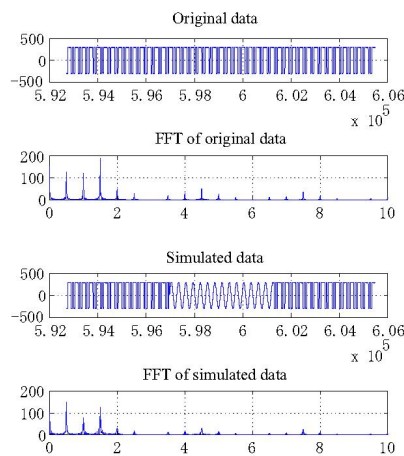

**Figure 8: Original data(above)and simulated data of waveform difference mutation(below).**

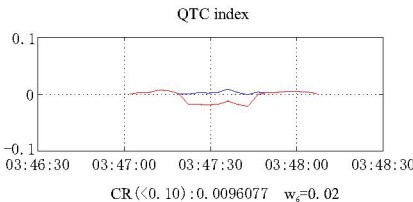

**Figure 9: QTC index of original data(blue)and simulated data of waveform difference mutation(red).**

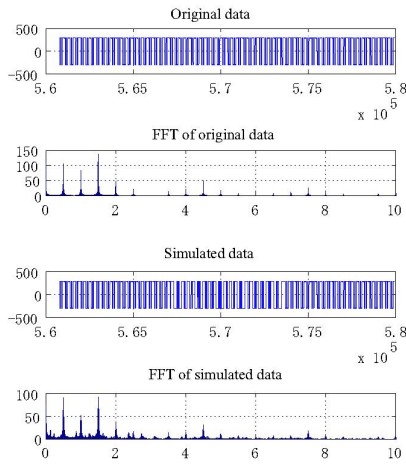

**Figure 10: Original data(above)and simulated data of waveform repetition mutation(below).**





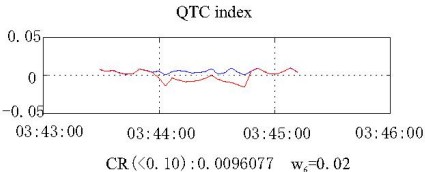

**Figure 11: QTC index of original data (blue) and simulated data of waveform repetition mutation (red).**

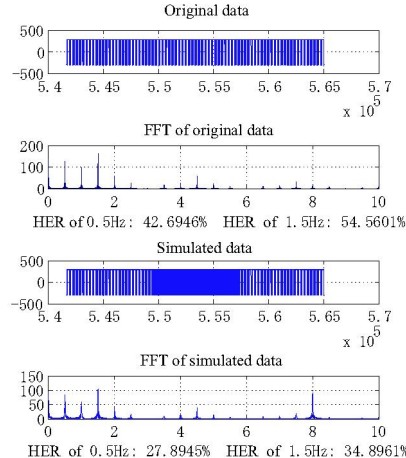

**Figure 12: Original data(above)and simulated data of harmonic energy ratio mutation(below).**

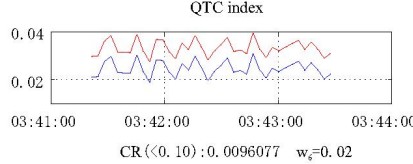

**Figure 13: QTC index of original data(blue)and simulated data of harmonic energy ratio mutation(red).**


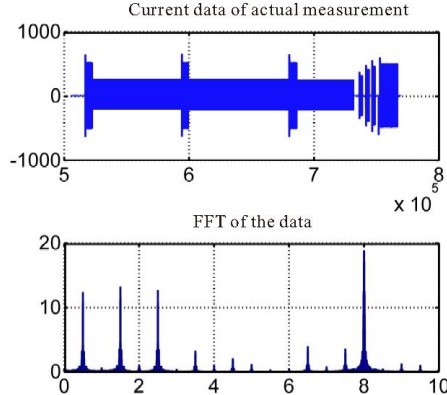

**Figure 14: The current data of actual measurement and FFT results.**

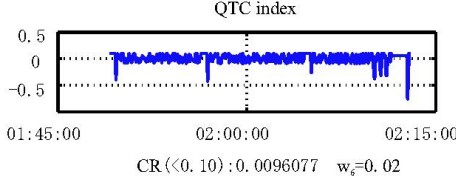

**Figure 15: QTC index of the actual measurement.**

**Table 1: Judgment matrix**

| $O$ | $a_i$ | $b_i$ | $c_i$ | $d_i$ | $e_i$ |
|---|---|---|---|---|---|
| $a_i$ | 1 | 1/3 | 1/3 | 5 | 3 |
| $b_i$ | 3 | 1 | 1 | 9 | 7 |
| $c_i$ | 3 | 1 | 1 | 9 | 7 |
| $d_i$ | 1/5 | 1/9 | 1/9 | 1 | 1/2 |
| $e_i$ | 1/3 | 1/7 | 1/7 | 2 | 1 |

**Table 2: RI for various n**

| $n$ | 1 | 2 | 3 | 4 | 5 | 6 | 7 | 8 | 9 |
|---|---|---|---|---|---|---|---|---|---|
| $RI$ | 0 | 0 | 0.58 | 0.90 | 1.12 | 1.24 | 1.32 | 1.41 | 1.45 |