# Peer review of "A comprehensive data quality evaluation method for the current of marine controlled-source electromagnetic transmitter based on Analytic Hierarchy Process"

_Geoscientific Instrumentation, Methods and Data Systems, 2019_

## Referee Comment (RC1) · Anonymous Referee #1 · 5 Sep 2019

Marine instruments for electromagnetic detection are well known to be difficult to make. One of the difficulties is the control of the transmitter, which pumps a huge amount of electrical current into the seawater. The quality of the output current need to be monitored the ensure the measured data are usable. The work presented in this paper proposes a new approach to do the quality control. I think it solved a very important issue in high-power marine electromagnetic instruments.

However, marine electromangetic technology is a very money-intensive and quite narrow subject. Not many researchers are in this area. To better attract a wider readership, I suggest the authors add some comments on how this new technique can be used in other instruments, for example, the EM transmitter desgined for land surveys or borehole geophysics.

I also think the computing time or the costs of running this method should be mentioned. It makes sure the new method is practical in the sea.

The authors used "mutation" for many times to refer to fluctuation or variation. Please check whether this usage is appropriate.

I am also curious if the introduction of the proposed method will change how a marine electromagnetic survey is carried out or how the data should be processed in comparision with the common practice.
* * *

---

## Short Comment (SC1) · 16 Sep 2019

This algorithm can be applied to other electromagnetic transmitters. The core of it is transmitting current data. For EM transmitter of land surveys or borehole geophysics, similar algorithms can also be made to control the quality. The corresponding QTC index can be calculated by inputting current data and corresponding parameters.And the calculation method is basically the same.
* * *
[Figure]

https://doi.org/10.5194/gi-2019-16, 2019.

---

## Author Comment (AC1) · 17 Sep 2019

Dear Anonymous Referee. Thank you very much for your appreciation of our work and for your precious comments on our manuscript. We reply the comments point by point in this letter, and corresponding changes will be made to improve the manuscript.

1) However, marine electromagnetic technology is a very money-intensive and quite narrow subject. Not many researchers are in this area. To better attract a wider readership, I suggest the authors add some comments on how this new technique can be

used in other instruments, for example, the EM transmitter designed for land surveys or borehole geophysics.

Reply: That's a very great suggestion. This algorithm can indeed be applied to other electromagnetic transmitters. The core of it is transmitting current data. For EM transmitter of land surveys or borehole geophysics, similar algorithms can also be made to control the quality. The corresponding QTC index can be calculated by inputting current data and corresponding parameters. And the calculation method is basically the same.

2) I also think the computing time or the costs of running this method should be mentioned. It makes sure the new method is practical in the sea.

Reply: Thank you for your kind reminding. The computing time of this method depends mainly on the number of each data block "N" and data length. Take Figure 3 in Section 4 as an example. The length of time is about 2 hours and 58 minutes and the sampling rate is 150Hz. The number of each data block is 32768. The calculation times of Fig.3a, Fig.3b and Fig.3c are 1.0410s, 1.4447s and 46.1602s respectively in MATLAB. And Fig.3a and Fig.3b are the most commonly used, so this computing time is acceptable in the sea.

3) The authors used "mutation" for many times to refer to fluctuation or variation. Please check whether this usage is appropriate.

Reply: The control of marine controlled source electromagnetic transmitter is usually on deck. Sometimes it is misoperated or some human factors lead to a sudden change of transmitting current. So, the word "mutation" used to emphasize these situations. However, I looked it up in the dictionary and found that the word is mostly used in genetics, which may not be suitable for use here. Thank you very much for your reminding. Now we have changed "mutation" to " variation".

4) I am also curious if the introduction of the proposed method will change how a

marine electromagnetic survey is carried out or how the data should be processed in comparision with the common practice.

Reply: As stated in the manuscript, the algorithm uses data from the current sensors of electromagnetic transmitter for comprehensive evaluation. Once we detect anomalous through the QTC index, we will immediately find out where the problem is and correct it in time. For example, frequency changes are usually not observed, but the QTC index can capture that. And then we check each attribute monitoring interface, quickly find out the fluctuation of frequency sub-curve, feedback to the host computer, adjust the transmitting frequency. For data processing, it's the same as the common practice.

---

## Referee Comment (RC2) · Axel Djanni (Referee) · 18 Oct 2019

The article presents a methodology to quantitively evaluate measured current in marine CSEM. Despite the quality of the results which seem promising, I have comments to make on the contents and the technical approach layout in this article. These should be addressed properly in order to publish it.

1- I know English is not the first language of the authors but I may suggest writing your article in the following tenses: **) As the subject of your sentence is mostly about the

study you have carried out, then you should use the present tense. **) Your conclusion and interpretation of the results should be written ONLY in the present tense.

Abstract: The first 4 lines should be removed. They don t give any new information that we don t know. I suggest starting with something like: "We present a QC methodology . . .". After ". . .within 2%.", I suggest starting with "The key findings are that . . .".

Introduction: There are typos errors that I can t go through each of them unfortunately. Please read it again! After, "Mittet et al., 2008". You stated that "there is no. . .". . .Are you sure about this affirmation? To make your point clear, I suggest starting the sentence with the name of the authors you are citing: for example – Edward, 2005 states that . . ..

Transmitting current analysis "The MCSEM operation data processing . . .", this sentence does not make sense. Can you please re-write it? Do you mean that the transmitting current quality influences the inversion?

Evaluation algorithm: Okay

Frequency stability: Please clarity what frequency you evaluate in ai. . .is it the fundamental or the harmonic?

Positive and negative amplitude: Can you please clarify how you obtain $b1 = 0.001$?? Also, it should be $b0$? Same for $c1$

Ideal waveform difference: In the first sentence, what d you mean by "single square wave frequency"? Did you mean: "fixed period"? Also, the computation of $di$ is a little problematic. . .it will always average to 0. Or am I wrong? What if the noise is correlated? I will suggest to compute the square roots of the output instead. The same observation goes to equation (6). Waveform repetition: I guess $b$ is the number of samples per period or? Please clarify.

Evaluation algorithm and comprehensive index: okay

Conclusion You should explain the ideal waveform your methodology works effectively and suggest the error one can have using other types of waveforms. Also, during the field trial, what device have you used to measure the current? How accurate is it?

Thanks very much for your contribution. I look forward to reading your feedback.

---

## Author Comment (AC2) · 6 Nov 2019

Dear Axel Djanni. Thank you very much for your comments and suggestions on the details of our manuscript, which have greatly improved the quality of our manuscript. We reply the comments point by point in this letter after our discussion and modification, the marked-up manuscript and reply with two pictures version are both in the supplement file.

1) 1- I know English is not the first language of the authors but I may suggest writing

your article in the following tenses: **) As the subject of your sentence is mostly about the study you have carried out, then you should use the present tense. **) Your conclusion and interpretation of the results should be written ONLY in the present tense.

Reply: Thank you very much for your suggestions on writing tenses. We checked the whole manuscript carefully and corrected the tense problem. The corresponding changes have been marked in the manuscript (Abstract.line_22, section2.line_22, section3.2.line_26, section3.6.line_24, section3.7.line_18, section4, section5 and conclusion). Thank you for your reminding. We will pay more attention to writing from now on.

2) Abstract: The first 4 lines should be removed. They don't give any new information that we don't know. I suggest starting with something like: "We present a QC methodology ...". After "...within 2%.", I suggest starting with "The key fi̧ndings are that ...".

Reply: This is a very good suggestion. We simplify the first several lines and start with "We present a QC methodology for the current of electromagnetic transmitter of marine controlled-source electromagnetic." And then change the end to "The key fi̧ndings are that the QTC index changes to more than 4% and some curvilinear features are observed if the transmitting current quality is poor. These results will provide a positive, significant guide for the evaluation and monitoring of transmitting current data in marine experiments."

3) Introduction: There are typos errors that I can t go through each of them unfortunately. Please read it again! After, "Mittet et al., 2008". You stated that "there is no..."...Are you sure about this affi̧rmation? To make your point clear, I suggest starting the sentence with the name of the authors you are citing: for example – Edward, 2005 states that ....

Reply: We checked these places carefully and there was no problem. Most of them are unrecognized names. After, "Mittet et al., 2008". What we meant was that we haven't

found a paper about current quality control yet. But maybe our translation is not good and the expression is not very accurate. We change the sentence to "Edwards, 2005, He, 2009 and Luan, 2018 states some data processing methods of controlled-source electromagnetic, but they do not make too much research on current quality." now.

4) Transmitting current analysis "The MCSEM operation data processing ...", this sentence does not make sense. Can you please re-write it? Do you mean that the transmitting current quality influences the inversion?

Reply: We are sorry that we didn't translate the meaning clearly. Yes, what we meant was that the transmitting current quality influences the data pre-processing and inversion. Now we rewrite it with "The MCSEM data pre-processing and inversion are influenced by the transmitting current quality.".

5) Frequency stability: Please clarity what frequency you evaluate in ai...is it the fundamental or the harmonic?

Reply: Both. Although most frequencies are useless for us, we have designed the algorithm that can analyze all frequencies. Because we want to calculate the frequency spectrum of QTC to compare the stability of different frequencies like Fig3(c). It mainly shows the changes in the QTC index with frequency and time. Blue indicates smaller stability values and more stable transmitting current, whereas red indicates the opposite.

6) Positive and negative amplitude: Can you please clarify how you obtain $b_1 = 0.001$?? Also, it should be $b_0$? Same for $c_1$

Reply: We're sorry we didn't explain $b_1$ clearly. It is the stability of the first data block according to the definition, but we design this parameter to identify changes, so the first data is meaningless. Then we give this initial value within 1%, which has no practical significance and do not affect the final overall data. This was not specifically stated in the previous manuscript, and we now add the explanation in section3.2 and section3.3.

Thank you very much for your reminding.

7) Ideal waveform difference: In thefirst sentence, what d you mean by "single square wave frequency"? Did you mean: "fixed period"? Also, the computation of di is a little problematic...it will always average to 0. Or am I wrong? What if the noise is correlated? I will suggest to compute the square roots of the output instead. The same observation goes to equation (6). Waveform repetition: I guess b is the number of samples per period or? Please clarify.

Reply: (a) In the first sentence, what d you mean by "single square wave frequency"? Did you mean: "fixed period"?

It means "square wave of a single frequency signal". We're trying to explain that we use two kinds of signals alternately in our experiments. Maybe our translation is not very accurate. Now we rewrite it with "square wave of a single frequency signal".

(b) Also, the computation of di is a little problematic...it will always average to 0. Or am I wrong? What if the noise is correlated? I will suggest to compute the square roots of the output instead. The same observation goes to equation (6)

Yes. The di is related to noise. So, it is almost impossible to be equal to 0. Now our transmitter is equipped with a channel to record ideal control waveform data. We can directly see the difference between the two by reading data, and di is to quantify it. Computing the square roots is a very good suggestion and we will consider this way. Thank you very much.

(c) Waveform repetition: I guess b is the number of samples per period or? Please clarify.

Yes, it is. We explain b in the last sentence of section3.5. "b is the number of each cycle of the transmitting waveform.". We use one or more transmitting waveform periods as one sample data. The b is the number of it. We rewrite it with "and b is the number of samples per cycle of transmitting waveform.".
8) Conclusion You should explain the ideal waveform your methodology works effectively and suggest the error one can have using other types of waveforms. Also, during the field trial, what device have you used to measure the current? How accurate is it?

Reply: We are not very clear what you mean here. Do you mean the error between ideal waveform and working waveform? Or the difference between good waveforms and poor waveforms? We explained this criterion in the conclusion. QTC indices less than 2% are normal and those more than 4% indicate current data is poor which require correction in a timely manner. The current sensor we used is a device manufactured in China. It is a closed-loop Hall current sensor, as shown in the picture in the supplement file. And its accuracy is $\pm0.4\%$ at $25°$C.

If there is anything we don't explain clearly, welcome to discuss. Thanks again for your kind reminding and very helpful suggestions.

Please also note the supplement to this comment:
https://www.geosci-instrum-method-data-syst-discuss.net/gi-2019-16/gi-2019-16-AC2-supplement.pdf

―――――――――――――

**Supplement:**

Reply to Axel Djanni:

Dear Axel Djanni. Thank you very much for your comments and suggestions on the details of our manuscript, which have greatly improved the quality of our manuscript. We reply the comments point by point in this letter after our discussion and modification, and the marked-up manuscript follows this reply.

1) **1- I know English is not the first language of the authors but I may suggest writing your article in the following tenses: \*\*) As the subject of your sentence is mostly about the study you have carried out, then you should use the present tense. \*\*) Your conclusion and interpretation of the results should be written ONLY in the present tense.**

Reply: Thank you very much for your suggestions on writing tenses. We checked the whole manuscript carefully and corrected the tense problem. The corresponding changes have been marked in the manuscript (Abstract.line_22, section2.line_22, section3.2.line_26, section3.6.line_24, section3.7.line_18, section4, section5 and conclusion). Thank you for your reminding. We will pay more attention to writing from now on.

2) **Abstract: The first 4 lines should be removed. They don't give any new information that we don't know. I suggest starting with something like: "We present a QC methodology ...". After "...within 2%.", I suggest starting with "The key findings are that ...".**

Reply: This is a very good suggestion. We simplify the first several lines and start with "We present a QC methodology for the current of electromagnetic transmitter of marine controlled-source electromagnetic." And then change the end to "The key findings are that the QTC index changes to more than 4% and some curvilinear features are observed if the transmitting current quality is poor. These results will provide a positive, significant guide for the evaluation and monitoring of transmitting current data in marine experiments."

3) **Introduction: There are typos errors that I can t go through each of them unfortunately. Please read it again! After, "Mittet et al., 2008". You stated that "there is no..."...Are you sure about this affirmation? To make your point clear, I suggest starting the sentence with the name of the authors you are citing: for example – Edward, 2005 states that ....**

Reply: We checked these places carefully and there was no problem. Most of them are unrecognized names. After, "Mittet et al., 2008". What we meant was that we haven't found a paper about current quality control yet. But maybe our translation is not good and the expression is not very accurate. We change the sentence to "Edwards, 2005, He, 2009 and Luan, 2018 states some data processing methods of controlled-source electromagnetic, but they do not make too much research on current quality." now.

4) **Transmitting current analysis "The MCSEM operation data processing ...", this sentence does not make sense. Can you please re-write it? Do you mean that the transmitting current quality influences the inversion?**

Reply: We are sorry that we didn't translate the meaning clearly. Yes, what we meant was that the transmitting current quality influences the data pre-processing and inversion. Now we rewrite it with "The MCSEM data pre-processing and inversion are influenced by the transmitting current quality.".

5) **Frequency stability: Please clarity what frequency you evaluate in ai...is it the fundamental or the harmonic?**

Reply: Both. Although most frequencies are useless for us, we have designed the algorithm that can analyze all frequencies. Because we want to calculate the frequency spectrum of QTC to compare the stability of different

frequencies like Fig3(c). It mainly shows the changes in the QTC index with frequency and time. Blue indicates smaller stability values and more stable transmitting current, whereas red indicates the opposite.

[Figure]

(c)

6) **Positive and negative amplitude: Can you please clarify how you obtain b1 = 0.001?? Also, it should be b0? Same for c1**

Reply: We're sorry we didn't explain b1 clearly. It is the stability of the first data block according to the definition, but we design this parameter to identify changes, so the first data is meaningless. Then we give this initial value within 1%, which has no practical significance and do not affect the final overall data. This was not specifically stated in the previous manuscript, and we now add the explanation in section3.2 and section3.3. Thank you very much for your reminding.

7) **Ideal waveform difference: In the first sentence, what d you mean by "single square wave frequency"? Did you mean: "fixed period"? Also, the computation of di is a little problematic...it will always average to 0. Or am I wrong? What if the noise is correlated? I will suggest to compute the square roots of the output instead. The same observation goes to equation (6). Waveform repetition: I guess b is the number of samples per period or? Please clarify.**

Reply: **(a) In the first sentence, what d you mean by "single square wave frequency"? Did you mean: "fixed period"?**
It means "square wave of a single frequency signal". We're trying to explain that we use two kinds of signals alternately in our experiments. Maybe our translation is not very accurate. Now we rewrite it with "square wave of a single frequency signal".

**(b) Also, the computation of di is a little problematic...it will always average to 0. Or am I wrong? What if the noise is correlated? I will suggest to compute the square roots of the output instead. The same observation goes to equation (6)**
Yes. The di is related to noise. So, it is almost impossible to be equal to 0. Now our transmitter is equipped with a channel to record ideal control waveform data. We can directly see the difference between the two by reading data, and di is to quantify it. Computing the square roots is a very good suggestion and we will consider this way. Thank you very much.

**(c) Waveform repetition: I guess b is the number of samples per period or? Please clarify.**

Yes, it is. We explain b in the last sentence of section3.5. "b is the number of each cycle of the transmitting waveform.". We use one or more transmitting waveform periods as one sample data. The b is the number of it. We rewrite it with "and b is the number of samples per cycle of transmitting waveform.".

8) **Conclusion You should explain the ideal waveform your methodology works effectively and suggest the error one can have using other types of waveforms. Also, during the field trial, what device have you used to measure the current? How accurate is it?**

Reply: We are not very clear what you mean here. Do you mean the error between ideal waveform and working waveform? Or the difference between good waveforms and poor waveforms? We explained this criterion in the conclusion. QTC indices less than 2% are normal and those more than 4% indicate current data is poor which require correction in a timely manner. The current sensor we used is a device manufactured in China. It is a closed-loop Hall current sensor, as shown in the picture below. And its accuracy is ±0.4% at 25℃.

[Figure]

If there is anything we don't explain clearly, welcome to discuss. Thanks again for your kind reminding and very helpful suggestions.

[revised manuscript text omitted]

删除了: mutation

[Figure]

**Figure 5: QTC index of original data(blue)and simulated data of frequency variation（red）.**

删除了: mutation

删除了: mutation

5  **Figure 6: Original data(above)and simulated data of amplitude variation(below).**

**Figure 7: QTC index of original data(blue)and simulated data of amplitude variation（red）.**

删除了: mutation

[Figure]

**Figure 8: Original data(above)and simulated data of waveform difference variation（below）.**

删除了: mutation

[Figure]

**Figure 9: QTC index of original data(blue)and simulated data of waveform difference variation（red）.**

删除了: mutation

[Figure]

**Figure 10: Original data(above)and simulated data of waveform repetition variation（below）.**

删除了: mutation

[Figure]

Figure 11: QTC index of original data (blue) and simulated data of waveform repetition variation（red）.

删除了: mutation

[Figure]

Figure 12: Original data(above) and simulated data of harmonic energy ratio variation（below）.

删除了: mutation

[Figure]

Figure 13: QTC index of original data(blue) and simulated data of harmonic energy ratio variation（red）.

删除了: mutation

[Figure]

**Figure 14: The current data of actual measurement and FFT results.**

[Figure]

**Figure 15: QTC index of the actual measurement.**

**Table 1: Judgment matrix**

| $O$ | $a_i$ | $b_i$ | $c_i$ | $d_i$ | $e_i$ |
|---|---|---|---|---|---|
| $a_i$ | 1 | 1/3 | 1/3 | 5 | 3 |
| $b_i$ | 3 | 1 | 1 | 9 | 7 |
| $c_i$ | 3 | 1 | 1 | 9 | 7 |
| $d_i$ | 1/5 | 1/9 | 1/9 | 1 | 1/2 |
| $e_i$ | 1/3 | 1/7 | 1/7 | 2 | 1 |

**Table 2: RI for various n**

| $n$ | 1 | 2 | 3 | 4 | 5 | 6 | 7 | 8 | 9 |
|---|---|---|---|---|---|---|---|---|---|
| $RI$ | 0 | 0 | 0.58 | 0.90 | 1.12 | 1.24 | 1.32 | 1.41 | 1.45 |